# Salvage of the 5-deoxyribose byproduct of radical SAM enzymes

Guillaume A.W. Beaudoin[1], Qiang Li[2], Jacob Folz[3], Oliver Fiehn [3,4], Justin L. Goodsell[2], Alexander Angerhofer[2], Steven D. Bruner[2] & Andrew D. Hanson [1]

5-Deoxyribose is formed from 5′-deoxyadenosine, a toxic byproduct of radical *S*-adenosylmethionine (SAM) enzymes. The degradative fate of 5-deoxyribose is unknown. Here, we define a salvage pathway for 5-deoxyribose in bacteria, consisting of phosphorylation, isomerization, and aldol cleavage steps. Analysis of bacterial genomes uncovers widespread, unassigned three-gene clusters specifying a putative kinase, isomerase, and sugar phosphate aldolase. We show that the enzymes encoded by the *Bacillus thuringiensis* cluster, acting together in vitro, convert 5-deoxyribose successively to 5-deoxyribose 1-phosphate, 5-deoxyribulose 1-phosphate, and dihydroxyacetone phosphate plus acetaldehyde. Deleting the isomerase decreases the 5-deoxyribulose 1-phosphate pool size, and deleting either the isomerase or the aldolase increases susceptibility to 5-deoxyribose. The substrate preference of the aldolase is unique among family members, and the X-ray structure reveals an unusual manganese-dependent enzyme. This work defines a salvage pathway for 5-deoxyribose, a near-universal metabolite.

[1] Horticultural Sciences Department, University of Florida, Gainesville, FL 32611, USA. [2] Department of Chemistry, University of Florida, Gainesville, FL 32611, USA. [3] NIH West Coast Metabolomics Center, UC Davis Genome Center, University of California Davis, Davis, CA 95616, USA. [4] Biochemistry Department, King Abdulaziz University, Jeddah 21589, Saudi Arabia. These authors contributed equally: Guillaume A. W. Beaudoin, Qiang Li. Correspondence and requests for materials should be addressed to S.D.B. (email: bruner@chem.ufl.edu) or to A.D.H. (email: adha@ufl.edu)

Radical *S*-adenosyl-L-methionine (SAM) enzymes occur in all domains of life and catalyze diverse reactions via the generation of highly reactive 5′-deoxyadenosyl radicals[1]. These enzymes function in key pathways that include the biosynthesis of thiamin, biotin, lipoate, and molybdopterin[1,2]. In the catalytic cycle of radical SAM enzymes, an intermediate 5′-deoxyadenosine radical abstracts a hydrogen atom from diverse substrates, forming 5′-deoxyadenosine (dAdo) as a byproduct[1]. If not removed, dAdo can reach toxic levels that inhibit radical SAM enzymes themselves[3–6]. Thus, dAdo buildup in *Escherichia coli* leads to growth-limiting deficiencies of biotin and lipoate[3].

dAdo is known to be converted to 5-deoxyribose or 5-deoxyribose 1-phosphate (dR1P) by a nucleosidase[3] or a phosphorylase[7], respectively. It has been shown in rats that dR1P is dephosphorylated and reduced to 5-deoxyribitol, which is excreted[8]. Because such excretion wastes a potentially valuable sugar, other disposal pathways that salvage 5-deoxyribose by recycling it to mainstream metabolites seem likely a priori to exist in nature. No 5-deoxyribose salvage pathways have yet been discovered, however.

It has been proposed that the archaeon *Methanocaldococcus jannaschii* metabolizes dAdo using enzymes similar to the 5′-methylthioadenosine phosphorylase and 5-methylthioribose 1-phosphate isomerase of the methionine salvage pathway, and that the 5-deoxyribulose 1-phosphate (dRu1P) so formed is converted to the aromatic amino acid precursor 6′-deoxy-5-ketofructose 1-phosphate[9]. We predicted instead that dRu1P simply undergoes aldol cleavage, yielding the central carbon metabolites dihydroxyacetone phosphate (DHAP) and acetaldehyde. The reaction sequence (Fig. 1a) would be analogous to the catabolic pathways of the 6-deoxyhexoses L-fucose and L-rhamnose in *E. coli*, which proceed through isomerase, kinase, and class II (metal-dependent) aldolase steps to give DHAP and lactaldehyde[10,11] (Supplementary Fig. 1a), and to the metabolism of 5′-fluorodeoxyadenosine to DHAP and fluoroacetaldehyde in *Streptomyces cattleya*[12] (Supplementary Fig. 1b). As in the suggested archaeal pathway[9], the first two reactions in our proposed pathway (Fig. 1a) mirror the first (kinase or phosphorylase) and second (isomerase) steps in 5-methylthioribose metabolism via the methionine salvage pathway[13,14].

In this study, we identified candidate genes specifying the proposed pathway (Fig. 1a) and confirmed that the corresponding enzymes have the predicted activities. In addition, we report the structure and mechanistic analysis of the key aldolase that characterizes the pathway. This work establishes a salvage pathway for disposal of 5-deoxyribose, a near-universal metabolite.

## Results

**Prediction of 5-deoxyribose salvage genes.** Based on the prediction that 5-deoxyribose salvage involves a kinase or phosphorylase, an isomerase, and a class II aldolase we searched prokaryote genomes for gene clusters encoding such enzyme trios using the SEED comparative genomics database and associated tools[15]. This analysis uncovered clusters encoding homologs of the methionine salvage enzymes 5-methylthioribose kinase or 5′-methylthioadenosine phosphorylase and 5-methylthioribose 1-phosphate isomerase, and of the fucose metabolism enzyme fuculose 1-phosphate aldolase (Fig. 1b). These three-gene clusters occur in six different bacterial phyla and are thus widely distributed (Fig. 1b).

Certain genomes having this cluster (e.g., *Myxococcus xanthus*, *Clostridium botulinum*) lack the other genes necessary for methionine salvage, and many *Bacillus* species have both these clusters and, elsewhere in the genome, a methionine salvage cluster with its own kinase and isomerase genes. These observations indicate that these three-gene clusters do not function in methionine salvage, and that the kinase, phosphorylase, and isomerase they encode are paralogs of methionine salvage enzymes. Consistent with this view, phylogenetic analysis splits the *Bacillus* kinase and isomerase into clades separate from the canonical methionine salvage enzymes (Supplementary Fig. 2a, b). The fuculose 1-phosphate aldolase homologs from the three-gene clusters and canonical fuculose 1-phosphate aldolases[16] likewise fall into separate clades (Supplementary Fig. 2c). Furthermore, fitness assays[17] show that *drdA* genes from diverse proteobacteria are not involved in the utilization of common sugars, which fits with a function in salvage of a sugar such as 5-deoxyribose.

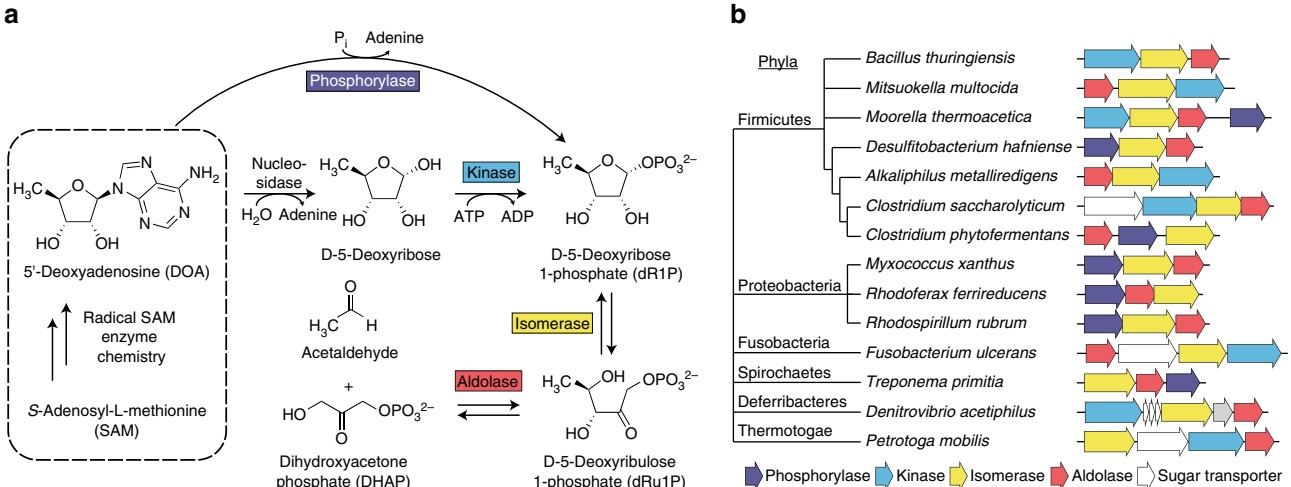

**Fig. 1** Potential routes of 5′-deoxyadenosine metabolism and gene clusters predicted to encode them. **a** 5′-Deoxyadenosine is converted to dR1P by a phosphorylase, or by a nucleosidase plus a kinase. dR1P is then metabolized to DHAP and acetaldehyde by the sequential action of an isomerase and an aldolase. **b** Gene clusters in genomes from six bacterial phyla that encode enzymes similar to 5-methylthioribose kinase or 5′-methylthioadenosine phosphorylase, 5-methylthioribose 1-phosphate isomerase, and fuculose 1-phosphate aldolase, a class II aldolase. Four of the clusters shown also include one or more putative sugar transporter genes. The sugar transporter genes in *Denitrovibrio acetiphilus* are not drawn to scale. The gray gene in *D. acetiphilus* is homologous to fucose mutarotase

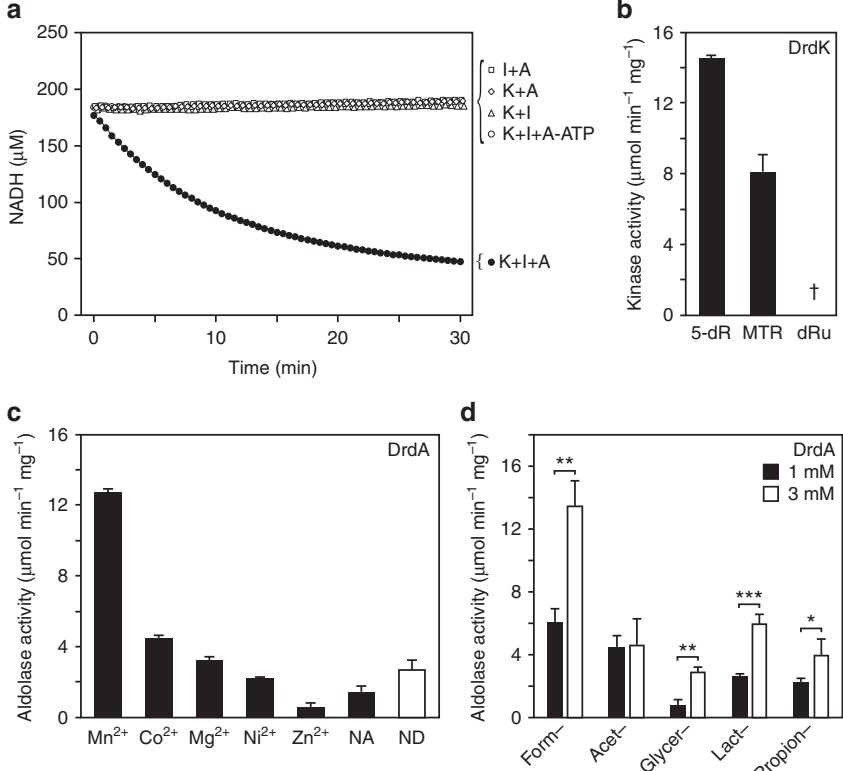

**Fig. 2** Activities of the *B. thuringiensis* kinase (DrdK), isomerase (DrdI), and aldolase (DrdA). **a** Coupled assay of DHAP formation from 5-deoxyribose when DrdK (K), DrdI (I), DrdA (A), and ATP are present in the reaction. DHAP formation was tracked via glycerol 3-phosphate dehydrogenase-catalyzed oxidation of NADH. Standard deviation for each point was <5 μM NADH. **b** DrdK is specific for 5-deoxyribose (5-dR) and 5-methylthioribose (MTR). Reactions contained 0.5 mM sugar and 1 mM ATP. Activity against 5-deoxyribulose (5-dRu) was undetectable (†), as was activity against 2-deoxyribose, ribose, fucose, fructose, fructose 6-phosphate, and glucose. Data are corrected for the ATPase activity of DrdK and are the mean of three replicates; error bars are the s.d. **c** DrdA prefers $Mn^{2+}$ or $Co^{2+}$ as cofactor. The enzyme was prepared from *E. coli* grown in unsupplemented LB, isolated by $Ni^{2+}$-affinity chromatography, desalted and dialyzed against EDTA to remove metals. Activity was then assayed without (NA) or with 0.1 mM metal chloride (or 2 mM for MgCl₂, to mimic intracellular levels). The activity of the undialyzed enzyme (ND) is also shown. Data are the mean of three replicates; error bars are the s.d. **d** The substrate preference of DrdA, measured by the loss of DHAP after 10 min from reactions containing 1 mM DHAP and 1 or 3 mM aldehyde (identified by prefix). Data are the mean of three (or for acetaldehyde six) replicates; error bars are the s.d. Significance was determined by *t*-test, $*P < 0.05$, $**P < 0.01$, $***P < 0.001$

**Biochemical validation of salvage activities**. We chose to investigate the kinase-isomerase-aldolase cluster from *Bacillus thuringiensis* (Fig. 1b) because this organism is genetically tractable. Each enzyme was expressed in *E. coli* with a hexa-histidine-tag and purified by $Ni^{2+}$ affinity chromatography (Supplementary Fig. 3a). Size exclusion chromatography indicated that the kinase and isomerase are dimers and the aldolase is an oligomer, possibly a tetramer like fuculose 1-phosphate aldolase[18] (Supplementary Fig. 3b, c).

The three enzymes, acting together in the presence of ATP, were necessary and sufficient to generate DHAP from 5-deoxyribose, as shown by using a coupled assay to measure DHAP formation (Fig. 2a). This result fits with the predicted reaction sequence (Fig. 1a) in which the kinase mediates ATP-dependent 1-phosphorylation of 5-deoxyribose, the isomerase converts dR1P to dRu1P, and the aldolase cleaves dRu1P to acetaldehyde and DHAP. We therefore propose the names 5-*d*eoxyribose *d*isposal (*drd*) kinase (*drdK*), isomerase (*drdI*), and aldolase (*drdA*) for the genes encoding these enzymes, and use these names from now on.

DrdK acted on 5-deoxyribose but not 5-deoxyribulose (Fig. 2b), confirming that phosphorylation precedes isomerization in the pathway. Of six other pentoses and hexoses tested, only 5-methylthioribose was a substrate (Fig. 2b), with a specificity constant ($k_{cat}/K_m$) close to that for 5-deoxyribose (Table 1).

This is consistent with the sequence similarity (59%) between DrdK and *Bacillus subtilis* 5-methylthioribose kinase[14]. The DrdK reaction product (Supplementary Fig. 4) is most likely the α anomer, as established for *B. subtilis* 5-methylthioribose kinase[19,20]. Additionally, DrdK had intrinsic ATPase activity (Supplementary Fig. 5 and Table 1), analogous to *B. subtilis* 5-methylthioribose kinase[21] and other kinases[22]. The specificity constant for DrdI with dR1P as substrate (Table 1) was comparable to that of a bacterial D-ribose isomerase[23] but substantially lower than that of *B. subtilis* 5-methylthioribose 1-phosphate isomerase[24].

Purified DrdA preparations were pink in color, suggesting a bound metal cofactor. The addition of $Mn^{2+}$ to demetallated DrdA resulted in higher aldolase activity than addition of $Co^{2+}$, $Mg^{2+}$, $Ni^{2+}$, or $Zn^{2+}$ (Fig. 2c); the apparent $K_{metal}$ value for $Mn^{2+}$ was $0.95 \pm 0.31$ μM (mean and s.d. of three determinations). $Zn^{2+}$ did not promote activity (Fig. 2c), which was unexpected because $Zn^{2+}$ is the native cofactor of fuculose- and rhamnulose 1-phosphate aldolases along with the majority of metal-dependent class II aldolases[25,26]. There is, however, one example of a homologous fructose-1,6-bisphosphate aldolase with preference for $Mn^{2+}$ in the aldol reaction[27].

Inductively coupled plasma mass spectrometry and EPR spectrometry were used to assign and characterize the bound metal in DrdA. To avoid metal binding to the hexa-histidine tag,

**Table 1 Kinetic characterization of the three 5-deoxyribose-metabolizing enzymes**

| Enzyme | Substrates | Co-substrates | $K_m$ (μM) | $k_{cat}$ (s$^{-1}$) | $k_{cat}/K_m$ (M$^{-1}$ s$^{-1}$) |
|---|---|---|---|---|---|
| DrdK (kinase) | 5-Deoxyribose | ATP | 178 ± 6 | 15.4 ± 0.4 | 8.7 ± 0.4 × 10$^4$ |
| | 5-Methylthioribose | ATP | 72 ± 20 | 6.2 ± 0.4 | 8.5 ± 2.4 × 10$^4$ |
| | ATP | 5-Deoxyribose | 216 ± 63 | 17.1 ± 2.5 | 7.9 ± 2.6 × 10$^4$ |
| | ATP (ATPase activity) | – | 84 ± 21 | 3.0 ± 0.3 | 3.6 ± 1.0 × 10$^4$ |
| DrdI (isomerase) | 5-Deoxyribose 1-P | – | 22 ± 7 × 10$^3$ | 3.5 ± 0.7 | 1.6 ± 0.6 × 10$^2$ |
| DrdA (aldolase) | 5-Deoxyribulose 1-P | – | 36 ± 3 | 6.3 ± 0.2 | 1.7 ± 0.1 × 10$^5$ |

The $k_{cat}$ values refer to one subunit. All values are means and s.d. for three replicate estimates

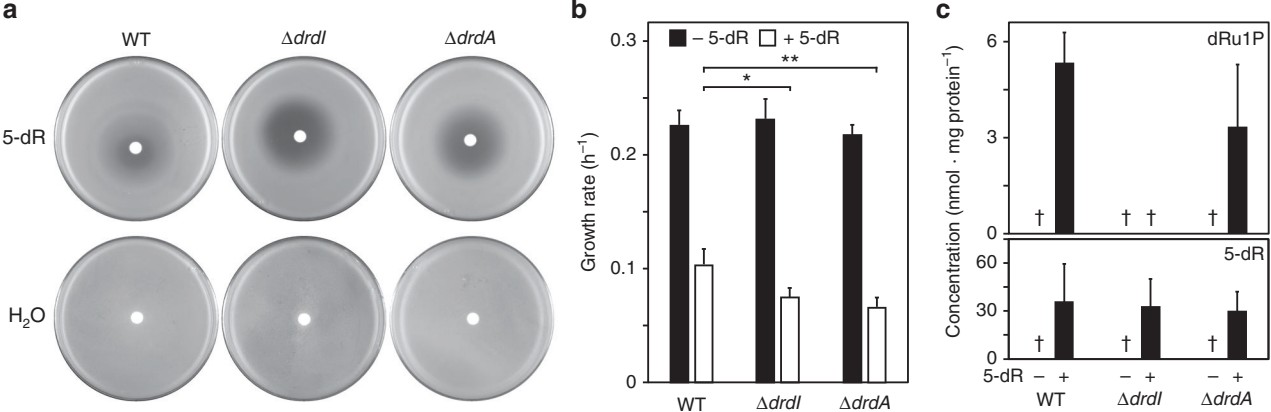

**Fig. 3** Growth and metabolic phenotypes of *drdA* and *drdI* deletant strains. **a** Deleting *drdA* or *drdI* exacerbates 5-deoxyribose toxicity. A disc containing 10 μL of 1 M 5-deoxyribose was applied to lawns of *B. thuringiensis* wild type (WT) or deletant cells. Note the larger zone of growth inhibition for the deletant strains. **b** Similarly, adding 1 mM 5-deoxyribose (5-dR) to liquid medium reduced the growth rate of the deletant strains significantly more than the wild type. Data are means of four replicates or three replicates for the Δ*drdI* strain; error bars are the s.d. Significance was determined by *t*-test, *$P < 0.05$, **$P < 0.01$. **c** Estimated levels of dRu1P and 5-dR in cells grown to an OD$_{600}$ of ~1.7, minus or plus 1 mM 5-dR. Data are means of 7–11 replicates; error bars are the s.d. †level below the limit of quantitation (~1.5 nmol mg protein$^{-1}$)

a tag-free version was prepared and purified using ammonium sulfate precipitation as the initial step. As the expression level of DrdA is high in the heterologous host *E. coli*, a mineral supplement was included in the growth medium in order to favor complete incorporation of less abundant metals. Without a supplement, zinc is the predominant metal found in the enzyme, however with an excess of metals included, manganese is incorporated at a higher percentage (Supplementary Fig. 6). These results are consistent with manganese being the preferred cofactor when low abundancy is excluded. In order to further characterize the Mn$^{2+}$-bound DrdA, EPR spectra were measured. Manganese-bound DrdA was prepared and purified before EPR analysis; the spectra (Supplementary Fig. 6) show six distinct peaks in the $g = 2$ region, which are broader than in the control lacking DrdA. This, along with the appearance of low-field bands near 1300 and 2000 G, is indicative of protein-bound Mn$^{2+}$[28]. No evidence of significant zero-field splitting was observed and there was no indication of a Mn$^{3+}$ species in the samples, consistent with the penchant for aldolases to bind and utilize divalent cations.

As with similar aldol cleavage reactions, dRu1P cleavage was reversible, enabling tests of substrate specificity by monitoring DHAP consumption in the presence of various aldehydes. Like fuculose- and rhamnulose 1-phosphate aldolases[29], DrdA accepted various aldehydes (Fig. 2d); acetaldehyde seems likely to be a preferred substrate in vivo because its saturating concentration is ≤1 mM, unlike the other aldehydes tested (Fig. 2d). The specificity constant for the dRu1P cleavage reaction (Table 1 and Supplementary Fig. 7) was comparable to those for the equivalent reactions of fuculose- and rhamnulose 1-phosphate

aldolases[11,30]. The stereochemistry at C3 and C4 of dRu1P prepared with DrdA was inferred to be 3R, 4R based on the diastereometric specificity of *E. coli* L-fuculose 1-phosphate aldolase[29] (Supplementary Fig. 8).

**Genetic evidence for the proposed pathway**. We created *B. thuringiensis* deletants for *drdI* and *drdA* to investigate 5-deoxyribose metabolism in vivo (Supplementary Fig. 9). Neither deletion impaired growth on solid nor liquid minimal medium alone (Fig. 3a, b). Addition of 5-deoxyribose inhibited growth of the wild type and both deletants, the effect being significantly greater in the deletants (Fig. 3a, b). These results show that 5-deoxyribose has toxic effects, and that *drdI* and *drdA* participate in a detoxification pathway.

The intracellular levels of 5-deoxyribose was below the limit of quantitation in all strains grown on minimal medium alone, but increased massively when 5-deoxyribose was supplied (Fig. 3c). The intracellular dRu1P level was likewise below the limit of quantitation in all strains grown on minimal medium, and increased in wild type cells supplied with 5-deoxyribose—but not in *drdI* deletant cells (Fig. 3c), which is consistent with the proposed role of the DrdI isomerase in 5-deoxyribose disposal (Fig. 1a). dRu1P also accumulated in *drdA* deletant cells (Fig. 3c), but not more than in wild type cells. Deleting *drdA* may not drive up dRu1P level beyond that in the wild type because the dRu1P concentration in wild type cells is clamped at a high level by the DrdA-mediated equilibrium between dRu1P and DHAP + acetaldehyde (Fig. 1a). Equilibrium constants for such aldol cleavage reactions[31,32] are typically ~10$^{-4}$, so that dRu1P formation is strongly favored.

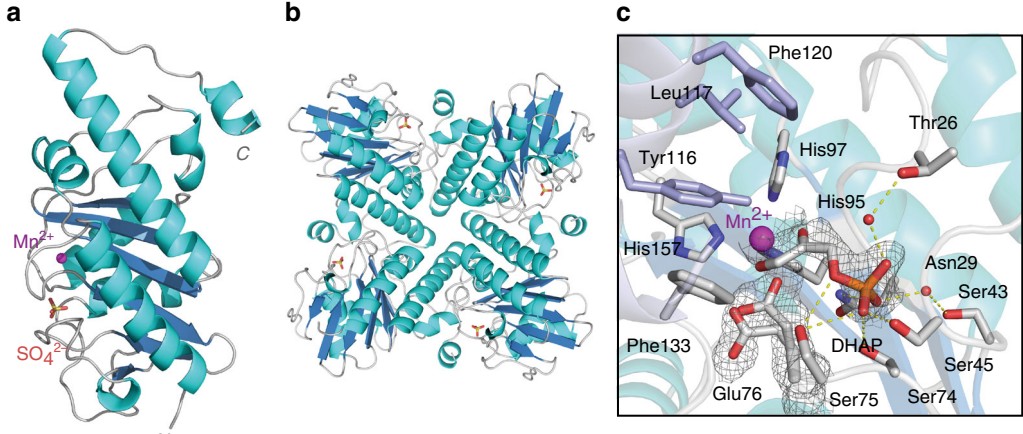

**Fig. 4** Structural basis for the 5′-deoxyadenosine pathway aldolase, DrdA. **a** X-ray crystal structure of DrdA (monomer) shown in ribbon representation with the active site manganese ion (purple) and sulfate anion (red) illustrated. **b** Structural representation of the tetrameric biological unit of the aldolase; the illustrated sulfate anion defines the active site. **c** View of the active site, illustrating amino acid side chains around the manganese ion. An electron density map is shown (calculated at $1.0\sigma$) around the bound substrate, dihydroxyacetone phosphate (DHAP), and the two positions of Glu76 in the co-complex structure

**Crystal structure and proposed mechanism of the aldolase DrdA.** Crystals of DrdA diffracted to a resolution of 1.55 Å in the space group $P\,42_12$ and the phase solution was determined by molecular replacement using fuculose 1-phosphate aldolase from *Streptococcus pneumoniae* (PDB entry 4C24, 53% sequence identity) as the search model[18]. The structure contains one monomer per asymmetric unit (Fig. 4a) and the crystal lattice provides a tetrameric complex (Fig. 4b), consistent with results from size exclusion chromatography and the proposed biological unit common to the class II aldolase family[18,33–35].

A structural homology search showed most significant homology to fuculose 1-phosphate aldolase, solved from various organisms, with homology to the *E. coli* enzyme the strongest[33,36]. The fuculose 1-phosphate aldolase family is a member of the class II aldolases, and as mentioned, uses a mononuclear $Zn^{2+}$ ion for Lewis acid-type catalysis. Fucose-processing pathways are present in many bacteria, notably in pathogens such as *S. pneumoniae*[18]. The sugar is commonly metabolized from mammalian and plant cell surface glycans and the aldolase step cleaves the sugar to DHAP and lactaldehyde (Supplementary Fig. 1a). Structures of fuculose-1-phosphate aldolase (FucA) determined from several sources including *S. pneumoniae* and *E. coli* exhibit a fold common to the class II aldolase family composed of a five-stranded antiparallel β-sheet core surrounded by eight α-helices of varying length. There is also structural homology to the L-ribulose-5-phosphate 4-epimerases, a similar family based on fold, but having alternate active site chemistry[34].

In order to gain structural insight into the basis for the substrate specificity of the aldol partners, we solved the co-complex structure of DHAP bound to DrdA. Attempts to solve the bound structures with acetaldehyde or the condensation product were not successful. The bound structure (Fig. 4c and Supplementary Fig. 10) shows the phosphate of DHAP occupying the position of sulfate in the apo-structure and bidentate coordination of the enediolate to $Mn^{2+}$.

The metal coordination around $Mn^{2+}$ is close to ideal tetrahedral in the apo-structure and is a distorted trigonal bipyramid in the complexed structure (Fig. 5a). In addition, the DHAP-bound structure defines the binding pocket of the aldehyde substrate. The observed larger pocket (Fig. 5b), as compared to FucA, is consistent with the diverse aldol acceptors

accommodated by DrdA as demonstrated biochemically. Analogous to *E. coli* FucA, binding of DHAP results in displacement of the coordinating glutamate residue (Glu76) by the enediolate of the co-substrate[30,36]. Glu76 rotates about the β/γ bond to accommodate the substrate and then is in position to act as an acid/base catalyst to facilitate the reaction. Overall, the evidence supports a mechanistic proposal (Fig. 5c) in which the displaced Glu76 acts as a base to initiate the aldol cleavage, generating a stable product structure and free acetaldehyde.

**Potential moonlighting connections to methionine salvage in bacteria and plants.** The similar specificity constants of DrdK with 5-deoxyribose or 5-methylthioribose as substrate (Table 1) suggest that this enzyme, and by extension the associated isomerase DrdI, might function in methionine salvage as well as 5-deoxyribose disposal. Supporting this possibility, *B. thuringiensis* DrdK, DrdI, and DrdA acting in tandem converted 5-methylthioribose to DHAP and mercaptoacetaldehyde (Supplementary Fig. 11). Comparative genomic evidence also supports this possibility. Thus, several bacteria (e.g., *Synechococcus* sp. WH7805, *Haliscomenobacter hydrossis*) do not have an isomerase gene in their methionine salvage gene cluster, but do have one that is clustered with *drdA* (Supplementary Fig. 12). Relatedly, in *Corallococcus coralloides*, the only predicted *drdA* gene is part of the methionine salvage cluster (Supplementary Fig. 12). On the other hand, the *Arabidopsis thaliana* genome encodes a single class II aldolase, fused to a HAD family phosphatase domain. The fusion enzyme (At5g53850, DEP1) catalyzes three successive steps in the methionine salvage pathway, the dehydration, enolization, and dephosphorylation of 5-methylthioribulose-1-phosphate[37]. Recombinant DEP1 showed aldolase activity towards 5-deoxyribulose-1-phosphate (Supplementary Fig. 13), suggesting that this activity, along with the kinase and isomerase involved in methionine salvage, could account for the metabolism of 5-deoxyribose in plants. Together, these results support the possibility that many organisms metabolize 5-deoxyribose via moonlighting activities rather than via dedicated enzymes.

**Discussion**

Many radical SAM enzymes form the byproduct dAdo, which is inhibitory[3–6] and consequently needs a disposal route[3,9]. Here we

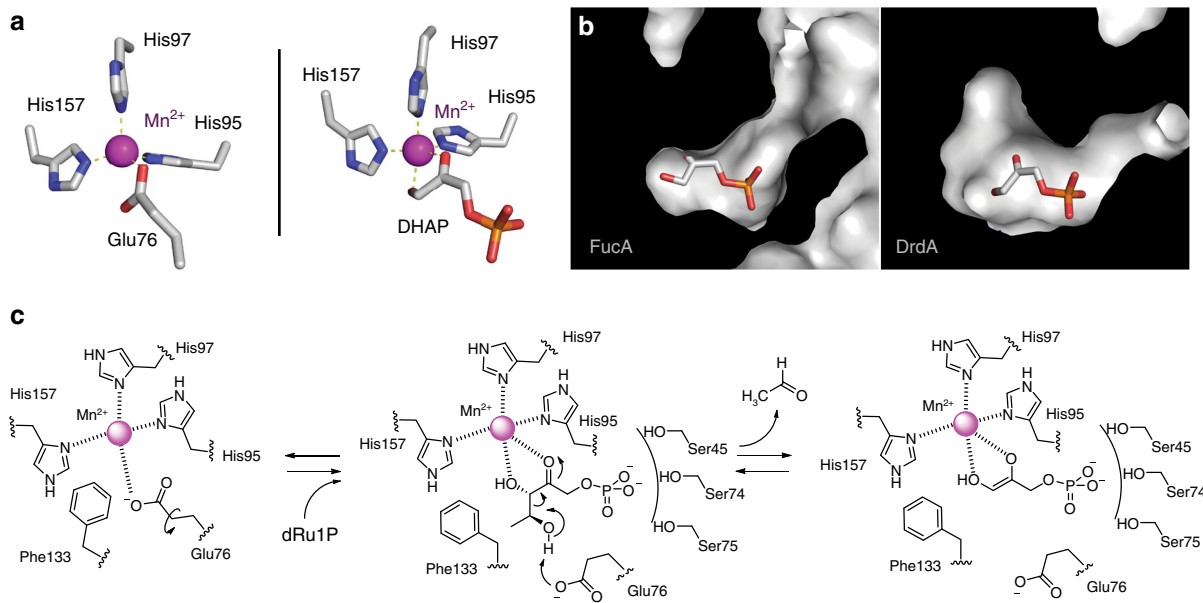

**Fig. 5** Substrate/metal binding and proposed enzymatic mechanism of DrdA. **a** Manganese binding coordination of unliganded and liganded DrdA; the substrate DHAP replaces the coordinating Glu76. **b** Active site surface representation showing the binding pocket for the acceptor, aldol substrate in DrdA and FucA (PDB code 4C24). **c** Mechanistic proposal for DrdA-catalyzed formation of the enediol intermediate along with release of acetaldehyde

demonstrate a dAdo disposal and salvage pathway leading via 5-deoxyribose, dR1P, and dRu1P to the mainstream metabolites acetaldehyde and DHAP (Fig. 1a).

The kinase, isomerase, and aldolase genes specifying this pathway are clustered in operonic arrangements in bacteria from several phyla (Fig. 1b). Clustering evidence also indicates that, in bacteria such as *Clostridium phytofermentans*, dR1P is formed directly from dAdo via a phosphorylase (Fig. 1b). The methionine salvage pathway can likewise start with either a kinase or a phosphorylase[14]. As the 5-deoxyribose disposal gene clusters in some bacteria include predicted sugar transporters (Fig. 1b), these organisms may salvage exogenous as well as endogenous 5-deoxyribose.

Although widespread, 5-deoxyribose disposal gene clusters are far from ubiquitous, implying that other disposal strategies exist. There are at least four possibilities. (i) The pathway proposed for *M. jannaschii*[9] in which enzymes similar to the phosphorylase and isomerase of the methionine salvage pathway generate dRu1P that is further metabolized to 6′-deoxy-5-ketofructose 1-phosphate. (ii) Excretion of 5-deoxyribose, analogous to the excretion of 5-methylthioribose by *E. coli*, whose methionine salvage pathway is incomplete[38]. (iii) Moonlighting by the enzymes of 6-deoxyhexose metabolism[10,11]. However, such enzymes are typically substrate-induced[39] whereas 5-deoxyribose formation is constitutive. (iv) Moonlighting by the methionine salvage kinase and isomerase, with an unidentified aldolase completing the pathway. Supporting this scenario, 5-methylthioribose kinase[40] and various sugar phosphate aldolases[27,41] are promiscuous, as is DrdA (Fig. 2d). Also, as noted above, the methionine salvage kinase or isomerase gene in some organisms clusters with an aldolase (Supplementary Fig. 12), which fits with dual roles for the kinase and isomerase, and the *Arabidopsis* trifunctional salvage enzyme DEP1 has DrdA activity (Supplementary Fig. 13).

It is important to note that inactivating the 5-deoxyribose disposal pathway in *B. thuringiensis* did not reduce growth rate unless 5-deoxyribose was added to the medium (Fig. 3b) because

this suggests that flux through the pathway is low under normal culture conditions. The observation that the 5-deoxyribose and dRu1P pools were too small to quantify in wild type and knockout cells not given 5-deoxyribose (Fig. 3c) likewise indicates low pathway flux. A low flux also fits with the relatively small number (15) of radical SAM enzymes encoded by *B. thuringiensis* genome compared to >40 in other Firmicutes[42], and with the functional annotations of these 15 enzymes. Three (pyruvate formate lyase activating enzyme and two coproporphyrinogen III oxidases) were probably inactive under the aerobic culture conditions used, seven are cofactors in low-flux cofactor synthesis pathways, and four mediate low-flux RNA modification reactions; the two others are of unknown function.

DrdA is a unique member of the class II aldolase family of enzymes, differing from most others in both the nature of the metal ($Mn^{2+}$) and the aldehyde co-substrate. Most members of the family contain an active site $Zn^{2+}$ acting as a Lewis acid-type catalyst. Although, like similar aldolases, DrdA is promiscuous with respect to the aldehyde co-substrate, it is unusual in preferring acetaldehyde. The established mechanism for class II aldolases involves coordination of the enediol intermediate on the bound metal and a conserved active site glutamate residue directs proton transfer of the aldol. Our two structures of DrdA, bound and unbound with substrate, along with previous data on class II aldolases, allow a detailed mechanistic proposal (Fig. 5c). The utilization of aldolases as chemoenzymatic tools is a well-established and useful approach to produce chiral intermediates. The diastereoselective DrdA provides an orthogonal tool both from a unique substrate, acetaldehyde, and a broad tolerance for alternate aldehyde acceptors.

Finally, it is worth noting that the puzzle of how bacteria remove the toxic byproduct of common radical SAM enzymes was easily solved by a dual comparative genomics/comparative biochemistry approach enabled by the SEED comparative genomics database[15]. This type of approach continues to grow in power as ever more genomes are sequenced[43].

## Methods

**Bioinformatics**. DNA and protein sequences were from GenBank or SEED[15]. Sequences were aligned with Clustal W[44]. Phylogenetic trees were constructed from Clustal W alignments by the Neighbor-joining method using MEGA5[45]. A representative set of >1300 bacterial and archaeal genomes was analyzed using SEED tools[15]. Results are encoded in SEED subsystem "5-Deoxyribose disposal" (http://pubseed.theseed.org/SubsysEditor.cgi?page=ShowSpreadsheet&subsystem=5-Deoxyribose_disposal).

**Chemicals**. Chemicals and reagents were from Sigma-Aldrich or Fisher Scientific except for 5-deoxyribose, from Ambinter (Orléans, France), BioMol Green reagent from Enzo Life Sciences (Farmingdale, NY), and 5-methylthioribose, which was prepared from methylthioadenosine as described[46].

***Escherichia coli* expression constructs**. Primers are given in Supplementary Table 1. Sequences were amplified from *B. thuringiensis* HD73-20 (BGSC 4D22) genomic DNA with Phusion High-Fidelity DNA polymerase (New England Bio-labs). *Arabidopsis* DEP1 was amplified from a cDNA clone (ARBC Stock #S64398). Amplicons were purified, digested with XbaI/XhoI (DrdK), NdeI/XhoI (DrdI), PciI/XhoI (DrdA) or NcoI/XhoI (DEP1) and ligated into pET28b (Novagen) previously digested with NheI/XhoI (DrdK), NdeI/XhoI (DrdI) or NcoI/XhoI (DrdA, DEP1). This generated N-terminal His$_6$-tags with a short linker for DrdK and DrdI and C-terminal His$_6$-tags for DrdA and DEP1. The N-terminally His$_6$-tagged *E. coli* fuculose 1-phosphate aldolase expression construct and strain were obtained from the ASKA collection[47].

**Production and purification of proteins**. Plasmids encoding *B. thuringiensis* DrdK, DrdI, DrdA, or *Arabidopsis* DEP1 were introduced into BL21 CodonPlus (DE3)-RIPL cells (Stratagene). *E. coli* fuculose 1-phosphate aldolase was expressed in AG1 (Stratagene) cells. Cultures (250 ml) were grown at 37 °C in LB medium with shaking at 300 rpm. When OD$_{600}$ reached 0.8, the temperature was lowered to 21 °C, IPTG was added (final concentration 1 mM), and cultures were incubated for a further 16 h. Cells were harvested by centrifugation (6000$g$, 10 min, 4 °C), resuspended in 50 mM sodium phosphate, 300 mM sodium chloride, pH 8.0, and sonicated. The lysate was centrifuged at 14,600$g$; imidazole was added (final concentration 10 mM) to the supernatant, which was incubated with 1 ml of HisPur™ Ni-NTA Superflow Agarose (ThermoFisher Scientific) slurry for 30 min at 4 °C. This was poured into a 13-cm polypropylene column (Evergreen Scientific) and allowed to drain. The column was washed with 50 ml of 50 mM sodium phosphate, 300 mM sodium chloride, 20 mM imidazole, pH 8.0 and proteins were eluted with 2 ml of this buffer supplemented with a further 230 mM imidazole, pH 8.0. The eluate was desalted on a PD-10 column (GE Healthcare) equilibrated with 50 mM HEPES-KOH, 50 mM KCl, pH 7.0, 10% glycerol. Protein solutions (typically 2–8 mg/ml) were then aliquoted, frozen in liquid N$_2$ and stored at −80 °C. To further purify *B. thuringiensis* DrdK to test for intrinsic ATPase activity, 1 mg was applied to a MonoQ 5/50 GL column (GE Healthcare), which was washed with 5 ml 50 mM HEPES-KOH, pH 7.5, 50 mM NaCl and eluted with a 5-ml, 0–1 M linear NaCl gradient, collecting 0.5-ml fractions. Eluate containing 100 μg of DrdK was applied to a Superdex 200 size exclusion column (GE Healthcare), which was eluted with 25 ml 50 mM HEPES-KOH, 50 mM KCl, pH 7.5, 10% glycerol. Fractions (1 ml) were collected. Purified DrdK was concentrated with Amicon Ultra-4 10K ultra-filtration columns (Millipore). Native molecular mass was estimated using a Superdex 200 column as above. Divalent cations were removed from DrdA and DEP1 preparations by adding 1 mM EDTA and holding at 4 °C for 16 h. EDTA was removed using a PD MiniTrap G-25 (GE Healthcare) desalting column equilibrated with 50 mM HEPES-KOH, 50 mM KCl, pH 7.5, 10% glycerol treated with 0.1% (w/w) Chelex 100 resin (Na$^+$ form) (Bio-Rad).

**Preparation of 5-deoxyribose 1-phosphate**. To 27.5 ml of 50 mM HEPES-KOH, 2 mM ATP magnesium salt, 2 mM MgCl$_2$, pH 7.5, containing 10 mg 5-deoxyribose (74.1 μmol) was added 2.5 ml of 3.2 mg/ml DrdK and the mixture was incubated for 4 h at 21 °C. Protein was removed with an Amicon Ultra-15 10K ultrafiltration column and the flow-through was applied to a 4-ml column of Dowex 1 × 8 (HCO$_3$⁻ form) (BioRad). The 5-deoxyribose 1-phosphate was eluted with a 30-ml, 0–1 M linear gradient of ammonium bicarbonate; 1-ml fractions were collected and those from the first phosphate ester-containing peak were pooled, lyophilized, redissolved in 10 ml water, and lyophilized again. This process was repeated until the weight remained constant (three times). A yellow oil (13.4 mg) was obtained and dissolved in 1 ml water; the solution was found to contain 48 mM 5-deoxyribose 1-phosphate (48 μmol, 64% yield, quantified via phosphate release following alkaline phosphatase treatment) and 8.7 mM free phosphate, and to be free of nucleosides.

**Preparation of 5-deoxyribulose 1-phosphate**. To 600 μl of 50 mM HEPES-KOH, 0.1 mM MnCl$_2$, pH 7.5 containing 5 mg DHAP lithium salt (25 μmol) was added 200 μl of 5.4 mg/ml DrdA; eight 20-μl additions of 2.0 M acetaldehyde were made at 15-min intervals and the reaction was then incubated for a further 1 h at 21 °C. Protein was removed by ultrafiltration as above; 0.33 ml of 27.5 mg/ml BaCl$_2$·2 H$_2$O (9.2 mg, 37.5 μmol) was added to the flow-through, followed by incubation

with shaking at 21 °C for 20 min. The precipitate was removed by centrifugation (14,800$g$, 10 min). Five volumes of absolute ethanol were added to the supernatant and the barium salt of dRu1P was left to precipitate for 16 h at −20 °C. The precipitate was harvested by centrifugation (14,800$g$, 10 min, 4 °C), washed with cold absolute ethanol, dried in air at 21 °C, and stored at −20 °C (2.9 μmol, 11.8% yield). Just before use, small amounts of the barium salt were suspended in 200 μl of 10 mM HEPES-KOH pH 7.5 and mixed with 10 μl of a 50% slurry of Dowex 50W-X8 (H$^+$ form) (BioRad). The mixture was vortexed for 30 s and centrifuged (21,000$g$, 10 s, 21 °C). The concentration of dRu1P in the supernatant was determined by a coupled assay with dRu1P, glycerol-3-phosphate dehydrogenase (GDH) and NADH.

**Preparation of 5-deoxyribulose**. To 200 μl of a solution of dRu1P prepared as above was added 1 μl (10 units) of calf intestinal phosphatase (New England Biolabs); the mixture was then incubated at 37 °C for 30 min. The phosphatase was removed by ultrafiltration as above. To determine the concentration of 5-deoxyribulose, the free phosphate produced was measured with the BioMol Green reagent.

**Enzyme assays**. Spectrophotometric assays were run at 21 °C in a Beckman DU 7400 spectrophotometer, monitoring NADH consumption by absorbance at 340 nm (A$_{340}$) every 10–15 s. 5-Deoxyribose kinase assays (100 μl) were basically as described[48]. The ADP generated by the kinase is converted to ATP by pyruvate kinase plus phosphoenolpyruvate, producing pyruvate, which is then reduced to lactate, consuming NADH. These assays contained 50 mM Tricine-HCl pH 8.0, 0.2 mM NADH, 1–1000 μM ATP magnesium salt, 2 mM MgCl$_2$, 1 unit of pyruvate kinase, 2 units of lactate dehydrogenase, 1 mM phosphoenolpyruvate, 0.5 mM 5-deoxyribose, and 0.25–0.5 μg kinase. To determine kinetic constants with respect to ATP, assays contained 500 μM 5-deoxyribose (kinase activity) or no 5-deoxyribose (ATPase activity). For all other assays, ATP concentration was 1 mM. The ATPase activity of the kinase was measured at successive purification steps using the same assay.

dR1P isomerase activity was measured using a coupled assay with DrdA and GDH. The dRu1P produced by the isomerase is cleaved by an excess of DrdA and gives rise to DHAP, which is reduced by GDH, consuming NADH. Assays (60 or 100 μl) contained 50 mM MOPS-KOH, pH 8.0, 0.2 mM NADH, 1 unit GDH, 20 μg DrdA, 0.01–30 mM dR1P, and isomerase. Assays with ≤4 mM dR1P had volumes of 100 μl and contained 1.4 μg isomerase. Assays with more ≥5 mM dR1P had volumes of 60 μl and contained 0.42 μg isomerase.

Assays to determine the metal dependence of the aldol cleavage reaction of DrdA (60 μl) were run in 50 mM HEPES-KOH, pH 8.2, 0.2 mM NADH, 1 unit GDH, 100 μM dRu1P, with 0.2–0.3 μg DrdA. Activities were measured without, or with the addition of 0.1 mM of metal chloride, with the exception of magnesium chloride, which was used at a concentration of 2 mM.

Assays to determine the kinetic parameters of the aldol cleavage reaction of DrdA (60 μl) were run in 50 mM HEPES-KOH, pH 8.2, 0.2 mM NADH, 1 unit GDH, 30 μM MnCl$_2$, 3–875 μM dRu1P, with 1 μg DrdA. Assays for the aldol addition reaction were discontinuous. Assays (100 μl) contained 50 mM HEPES-KOH, pH 8.2, 0.2 mM NADH, 30 μM MnCl$_2$, 1 mM DHAP, 1–3 mM aldehyde and 0.44 μg DrdA. Reactions were stopped after 11 min by centrifugation in an Amicon Ultra-0.5 10K ultrafiltration column (21,000$g$, 1 min, 21 °C). A 15-μl sample of the flow-through was added to 85 μl of 50 mM Tricine-HCl, 0.2 mM NADH, 1 unit GDH to determine the quantity of DHAP remaining. $K_m$ estimation was not attempted due to the inherent imprecision in stopping reactions by ultrafiltration.

Assays containing all three *B. thuringiensis* enzymes were run in 50 mM Tricine-HCl, pH 8.0, 0.2 mM NADH, 3 mM ATP magnesium salt, 2 mM MgCl$_2$, 1 unit GDH, 500 μM 5-deoxyribose, 10 μg 5-deoxyribose kinase, 10 μg dR1P isomerase, and 10 μg dRu1P aldolase.

*E. coli* fuculose 1-phosphate aldolase was assayed spectrophotometrically in 50 mM HEPES-KOH pH 7.3, 0.2 mM NADH, 1 unit GDH, 0.1 mM ZnCl$_2$, 4 μg EcFucA, 90 μM dRu1P. AtDEP1 was assayed spectrophotometrically in 50 mM HEPES-KOH pH 8.2, 0.2 mM NADH, 1 unit GDH, 500 μM dRu1P, with 25 μg AtDEP1.

**Crystallization of *B. thuringiensis* aldolase DrdA**. A tag-free variant of the *B. thuringiensis* aldolase was prepared by amplifying the gene using primers given in Supplementary Table 1; a stop codon preceded the XhoI site. The amplicon was cleaved with the matching restriction enzymes and ligated into pET-28a. The resulting expression plasmid was introduced into *E. coli* BL21(DE3) cells, which were then grown at 37 °C in LB medium containing 1% (v/v) trace mineral supplement (Trace Mineral Supplement, ATCC); when OD$_{600}$ reached 0.6, expression was initiated by adding IPTG (0.25 mM final concentration). Incubation was continued for 20 h at 16 °C; cells were then harvested by centrifugation, resuspended in 30 ml of 500 mM NaCl, 50 mM HEPES-KOH, pH 7.0 (lysis buffer), and lysed at 14,000 psi in a nitrogen-pressure microfluidizer cell (M-110L Pneumatic). The lysate was clarified by centrifugation, 15,000$g$ for 20 min at 4 °C, then purified by ammonium sulfate precipitation. (NH$_4$)$_2$SO$_4$ was added to the lysate to a final concentration of 75% saturation and stirred at 4 °C. Precipitated protein was collected by centrifugation (15,000$g$ for 20 min at 4 °C), then redissolved in 5 ml of

lysis buffer. The solution was dialyzed into low-salt buffer (100 mM NaCl, 50 mM HEPES-KOH pH, 7.0) and further purified by anion exchange chromatography (MonoQ HR 10/10, GE Healthcare) with a linear gradient of 0–500 mM NaCl over 30 min, followed by size-exclusion chromatography (Superdex 200) with 100 mM NaCl, 1 mM 2-mercaptoethanol and 50 mM HEPES-KOH, pH 7.0 and concentrated to ~6 mg/ml for crystallization trials.

Initial crystal screening was performed in a vapor-diffusion sitting-drop format using several commercial sparse matrix screens. Small clusters of needle crystals were identified in 200 mM NaCl, 25% (w/v) PEG-3350, 100 mM HEPES-KOH, pH 7.5. Optimization of salt and pH along with microseeding were carried out in hanging-drop format at 20 °C. The resultant crystals with a maximum size of ~30 × 200 × 30 μm$^3$ were obtained in a final condition that contained 200 mM NaCl, 28% (v/v) PEG-3350 and 100 mM HEPES-KOH, pH 7.5. Co-crystallization was performed by mixing with 20 equivalents of DHAP (final aldolase concentration = 6.6 mg/ml) and allowing the complex to equilibrate at 4 °C for 1 h. The resultant crystals with a maximum size of ~30 × 100 × 30 μm$^3$ were obtained in a final condition that contained 200 mM NaCl, 26% (w/v) PEG-3350 and 100 mM HEPES-KOH, pH 7.5.

**Data collection, processing, and structure refinement**. Diffraction data were collected on beamline 21-ID-G of the Life Sciences Collaborative Access Team (LS-CAT) facility, Argonne National Laboratory Advanced Photon Source (APS-ANL) at a wavelength of 0.9786 Å. Data were collected at 100 K, integrated, merged and scaled using the XDS package[49] to a resolution of 1.55 Å in space group $P42_12$ with one monomer per asymmetric unit. L-Fuculose 1-phosphate aldolase from *S. pneumoniae* (PDB entry 4C24, 53% sequence identity) was used as the search model for molecular replacement in Phaser[50]. PHENIX.AUTOBUILD succeeded in placing ~90% of the residues and the remaining residues were built into the maps manually using COOT[51]. Refinement was performed in PHENIX.REFINE and REFMAC5[52]. Sigma-A weighted, simulated annealing composite omit maps were used to judge and verify structures throughout refinement. Crystallographic data and refinement statistics are shown in Supplementary Table 2. Structural illustrations were prepared with PyMOL.

**Metal analysis**. Purified His-tag-free protein (grown in LB media with 1% (v/v) mineral supplement (Trace Mineral Supplement, ATCC) incorporated into the growth medium) was dialyzed for 16 h against 1 l of metal-free 0.1 M ammonium acetate, pH 6.5 that had been pretreated with 5% (w/v) Chelex 100 Resin, Na$^+$ form, then diluted with the same buffer to 10 ml (2.6 mg/ml final concentration). Concentrated nitric acid (trace metal grade) was then added to a final concentration of 2%. A blank 10 ml reference was prepared in parallel using identical procedures. Samples were analyzed by inductively coupled plasma mass spectrometry at the Center for Applied Isotope Studies of the University of Georgia (Athens, GA). Mn, Zn Cu, Mg, Ni, Co, and Fe were analyzed in each sample and the blank.

**EPR spectroscopy analysis**. The protein sample (as prepared for metal analysis, described above) was incubated with 1 mM EDTA at 4 °C for 16 h. A desalting column (Zeba Spin Desalting Columns 40K, ThermoFisher Scientific) was centrifuged at 1000g for 2 min at 4 °C to remove storage solution and washed thrice with storage buffer consisting of 100 mM NaCl, 1 mM 2-mercaptoethanol and 50 mM HEPES-KOH, pH 7.0, pre-equilibrated with 0.1% (w/v) Chelex 100 resin, Na$^{2+}$ form. The sample was slowly applied to the center of the resin bed and eluted with 900 μl of Chelex-treated storage buffer. Metal-free aldolase was collected by centrifuging at 3000g for 2 min at 4 °C and concentrated to 35 mg/ml. The solution was dialyzed against 100 mM NaCl, 1 mM 2-mercaptoethanol, 50 mM HEPES-KOH, pH 7.0 containing 1 mM MnCl$_2$. EPR spectra were recorded on a Bruker ELEXSYS-II E500 spectrometer fitted with a Bruker 4116DM dual mode resonator. Sample temperature was maintained at 5 K using an Oxford Instruments ESR900 continuous flow helium cryostat. Samples (200 μl) contained 35 mg protein/ml. Instrument parameters were as follows: 9.643431 GHz, 3550 Gauss center field, 7000 field sweep range, 40 ms conversion time, 10 Gauss modulation amplitude, 100 kHz modulation frequency, 7001 pts, 28 dB microwave attenuation (0.3170 mW). Parallel-mode X-band EPR was used to test for the presence of Mn(III) at 5 K.

**B. thuringiensis deletion constructs**. To delete *drdA*, amplicons were generated for the erythromycin cassette (primers 13 and 14), the upstream gene *drdI* primers (primers 11 and 12) and the downstream gene primers (primers 15 and 16). These were combined using Splicing by Overlap Extension (SOEing)[53], adding XbaI sites at the ends (primers 17 and 18). This amplicon was then digested with XbaI and cloned in the temperature-sensitive plasmid pDR244 (BGSC Accession 1A1133) that had been digested with XbaI to remove the *cre* gene. The *drdI* deletant was made in a similar way, with the erythromycin cassette (primers 21 and 22) combined with the upstream gene (primers 19 and 20) and downstream gene (primers 23 and 24) by SOEing (primers 25 and 26). This amplicon was then inserted into pDR244 as above.

**B. thuringiensis transformation and selection of deletants**. *B. thuringiensis* HD73-20 was transformed basically as described[54]. A 10-ml aliquot of BHIG (Brain Heart Infusion, 0.5% (v/v) glycerol) in a 250 ml baffled Erlenmeyer flask was inoculated with *B. thuringiensis* and incubated at 30 °C at 225 rpm for 16 h. A 5-ml portion was then diluted into 95 ml of pre-warmed BHIG and incubated at 30 °C at 300 rpm for 1 h. Cells were pelleted by centrifugation (5000g, 10 min, 4 °C). The pellet was resuspended in sterile, ice-cold 0.625 M sucrose, 1 mM MgCl$_2$, centrifuged again, resuspended in 10 ml of the same solution and 400 μl of the suspension was combined with 10 μl of *dam* and *dcm* methylation-free plasmid DNA (5 μg) isolated from *E. coli* INV110 (Invitrogen) and added to an ice-cold 0.4 cm electroporation cuvette. The cuvette was pulsed with a BioRad GenePulser with settings of 2.5 kV, 25 μF, 200 Ω and the sample was then diluted with 1.6 ml BHIG, incubated at 30 °C for 3 h, plated on LB-agar supplemented with 1 μg/ml erythromycin, 25 μg/ml lincomycin, 100 μg/ml spectinomycin and incubated at 30 °C for 20 h. To cure the plasmid and select for deletants, single colonies were streaked on LB-agar supplemented with 1 μg/ml erythromycin, 25 μg/ml lincomycin and incubated at 37 °C. Survivors were screened on LB-agar at 30 °C for sensitivity to spectinomycin. Spectinomycin-sensitive colonies were grown in 2 ml LB supplemented with 1 μg/ml erythromycin, 25 μg/ml lincomycin for 16 h. Genomic DNA was extracted using the GeneJET Genomic DNA Purification Kit (ThermoFisher Scientific). The position of the antibiotic cassette and absence of the wild-type gene were verified by PCR.

**B. thuringiensis growth for metabolomics experiments**. Glassware for *B. thuringiensis* grown in ED medium[55] was treated with Rain-X to prevent adhesion of the bacteria to vessel walls. Single colonies of the wild-type, *drdA*, and *drdI* deletant strains were inoculated in 2 ml of ED medium in 13-ml glass test tubes and grown for 48 h at 30 °C at 220 rpm. These cultures were used to inoculate (to an OD$_{600}$ of 0.05) 5 ml of ED medium in 25-ml Erlenmeyer flasks, which were incubated at 30 °C at 220 rpm until they reached OD 1.5–1.8. Cells were then harvested by centrifugation (21,000g, 15 s, 21 °C), flash frozen in liquid N$_2$ and stored at −80 °C.

**Metabolomics analysis**. Cell pellets of wild-type and *drdA* and *drdI* deletant strains were thawed on ice and resuspended in 0.5 ml acetonitrile:water (80:20 v/v) in 1.5 ml Eppendorf tubes. One half gram of 1.6 mm stainless steel balls were added to each tube and processed in a Geno/Grinder (SPEX) for 1 min at 1500 rpm. Each sample was shaken on an orbital shaker at 4 °C for 5 min at maximum speed, centrifuged (2 min, 14,000g), and 0.5 ml supernatant was transferred to a clean Eppendorf tube. The extraction was repeated, for a total extract volume of 1 ml. In a clean 1.5 ml Eppendorf tube, 475 μl of extract was dried under vacuum overnight and stored at −20 °C for no more than 1 week before derivatization and GC TOF-MS analysis. Standard solutions of 5-deoxyribose, dR1P, and dRu1P were diluted in acetonitrile:water (80:20) and dried as above. Samples and standards were analyzed as described[56]. Briefly, dried extract was redissolved in 10 μl of 40 mg/ml *O*-methoxyamine hydrochloride in pyridine and shaken at 30 °C for 1.5 h. To each tube 90 μl of *N*-methyl-*N*-(trimethylsilyl)-trifluoroacetamide including fatty acid methyl ester retention index markers was added and shaken at 37 °C for 30 min. Within 48 h of derivatization all samples were injected on an Agilent 6890 gas chromatograph coupled to a Leco Pegasus III time-of-flight mass spectrometer. A Restek RTX-5Sil MS (95% dimethyl/5% diphenyl polysiloxane) column with 30 m length, 0.25 mm i.d., and 0.25 μm film thickness was used with a 10 m guard column. Data was acquired from 85 to 500 *m/z* at 17 spectra/s and 1850 V detector voltage. Peaks were deconvoluted and detected using Leco ChromaTOF software and matched to FiehnLib mass spectra and retention time library. Binbase software was used for post-curation and peak replacements. The sum of intensities for all known compounds was used to normalize data. The 5-deoxyribose and dR1P contents of samples were determined using authentic standards. The protein content of *B. thuringiensis* cells was determined by the BCA method (ThermoFisher Scientific) and found to be 145 ± 15 μg per ml of culture at OD 1. Metabolite levels were expressed per mg of protein.

**Data availability**. Coordinates and structure factors for the aldolase structures have been deposited in the RCSB Protein Data Bank under PDB accession codes 6BTD (DrdA) and 6BTG (DrdA+DHAP). Other datasets generated and analyzed during the current study are available from the corresponding authors on request.

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

## Acknowledgements

This work was supported by the U.S. National Science Foundation grants MCB-1611711 (to A.D.H. and S.D.B.), by MCB-1611846 (to O.F.), and by the C.V. Griffin Sr. Foundation. We thank Daniel Ziegler of the Bacillus Genetic Stock Center, Alisdair Boraston, Valérie de Crécy-Lagard, and Eduardo García-Junceda for bacterial strains and clones.

## Author contributions

A.D.H. and S.D.B. devised the project. G.A.W.B. and A.D.H. carried out comparative genomic and biochemical analyses. J.F. and O.F. ran metabolomics analyses. J.L.G. and A.

A. carried out EPR analyses. Q.L. and S.D.B. performed crystallographic studies. A.D.H., G.A.W.B., and S.D.B. wrote the manuscript.

## Additional information

**Competing interests:** The authors declare no competing interests.

