## [Peer Review File · Nature Communications]

Reviewers' comments:

Reviewer #1 (Remarks to the Author):

Despite its importance metabolism is still considered as a side-component of what matters about what life is. Recently several papers tried to stress that there was an underlying logic behind what metabolism and why it matters, and the laboratory of the last author of this work has played a remarkable role in putting the picture straight.

The present paper is very interesting, as it unravels the fate of widely spread metabolite, 5'-deoxyadenosine. I have just a question, linked to the fact that the authors did not explore here the phylogeny of the enzymes they identified. Kinase DrdK is likely to belong to a large family of enzymes. It would be interesting to get some discriminating features of the enzyme in particular contrasting it with 5-methylthioribose kinase. Indeed, in *B. cereus* for example there are two highly similar enzymes, one that can phosphorylate 5-methylthioribose, and the other one that presumably phosphorylates 5-deoxyribose, both belonging to the choline kinase family (PMID: 11545674). This may allow extraction of a signature that would help annotating further genomes.

Indeed, an important feature of the work is that identification of relevant genes and enzyme activities will contribute to improve genome annotations (provided somebody cares to transfer the relevant annotations at the right places so that automatic pipelines could discover their importance). The work combines genetics and biochemistry and the background reasoning is fairly easy to follow. I have therefore no particular suggestions for improvement.

However, precisely because this work should be widely used by annotation pipelines, I think that there could be some improvement in the nomenclature. I am well aware, for example, that S-adenosylmethionine is commonly abbreviated into SAM. This is however unfortunate, for many reasons. In particular software for automatic retrieval, when using SAM as a keyword will stumble on so many entries that this will become useless. The common abbreviation (accepted by IUPAC) AdoMet would help considerably in that matter. The same view holds for several other abbreviations in the paper. DOA could be dAdo (convenient but still somewhat confusing). Drd might be OK (some overlap with *Drosophila* proteins).

Reviewer #2 (Remarks to the Author):

The manuscript by Beaudoin et. al. describes the characterization of novel salvage pathway for 5'-deoxyadenosine (DOA). The 5'-deoxyadenosyl radical is produced by the reductive cleavage of S-adenosyl-L-methionine by Radical SAM (RS) enzymes, which in turn, utilize it to carry out chemistry producing DOA. It has been shown by Cronan and coworkers that the accumulation of DOA in *E. coli* inhibits growth. Further, this phenotype was shown to be a direct inhibition of the RS enzymes BioB and LipA. Since that time, it has also been shown that DOA inhibits many more RS enzymes under *in vitro* assay conditions. It is well known that several enzymes will "moonlight" and cleave DOA into 5'-deoxyribose and adenine, such as MTA and SAH nucleosidases, but it has been a long-standing question as to the ultimate fate of 5'-deoxyribose. Beaudoin et. al. describe the characterization of three enzymes that sequentially convert 5'-deoxyribose first to 5-deoxyribose 1-phosphate, then 5-deoxyribulose 1-phosphate, and finally to DHAP and acetaldehyde—common central carbon metabolites. In addition, they describe the crystal structure of DrdA, the manganese-dependent aldolase in this pathway. Overall, the work is high quality and the paper is well written. However, one of the major issues with the paper is the data in Fig. 3. It worries me that the intercellular concentration of 5-dR has to reach 20-40 mM to see any defect in growth rate with both the Wt or mutant strains of *B. thuringiensis*. I worry that DrdI, DrdK, and DrdA are only "moonlighting" *in vitro*, and a different set of enzymes are responsible for the removal of 5-dR *in vivo*. In fact these enzymes seem more effective at removing 5-methylthioribose (Supp Fig. 10). This and the following issues should be addressed before publication.

Major comments:

Online Materials and Methods should contain a section describing how DrdA was prepared in the presence of added metals for ICP analysis. It states in the paper that “a mineral supplement was included in the growth medium,” but is not described in the Methods. Also, the metal dependent assays are briefly described in the Fig. 2 legend, but not in the Methods. Was the enzyme used from mineral supplement growths plus added metals or just from LB growth and then added metals? Is the same true for the metal analysis? These two experiments should be congruent. These issues make it difficult to evaluate the metal dependence of DrdA.

Supplementary Fig. 6a should be converted to metal/protein monomer. The numbers have little value as presented. In panel b the authors present an EPR spectra of their protein. They state in the paper “ICP- MS and EPR spectrometry were used to identify the bound metal in DrdA.” In the methods section, the authors don't say how much protein was used, but added 1 mM MnCl₂: no other metals are listed. This is a very biased experiment. In addition, Mn⁺² always has a six-line spectrum whether bound or free in solution. Based on the structure of DrdA, the Mn is bound by three His residues. These residues should contribute hyperfine coupling to the EPR spectra, which should be present and can be simulated. Therefore the EPR spectra needs better characterization. Also, the as-isolated protein (without added metals) from the mineral supplement media seems like it would be a better source for the sample. This should be done and would definitively prove the Mn is bound to the enzyme as the structure predicts.

Minor comments:

Reference 6 is to an enzyme called ThiC. Although the enzyme produces DOA, it is not a member of the RS superfamily of enzymes. The authors state this mistake in the sentence “If not removed, DOA can reach toxic levels that inhibit radical SAM enzymes themselves” by referencing ThiC as an RS enzyme.

First sentence of bottom paragraph on Page 2: “It has been proposed, but not proved, that the archaeon” seems redundant. “But not proved” should be removed.

It should be discussed more about what happens in species like *E. coli*. *E. coli* has 20 RS enzymes and several of them have been shown to be particularly sensitive to inhibition by DOA both *in vivo* and *in vitro*.

Title: Salvage of the 5-deoxyribose byproduct of radical SAM enzymes

Response to Reviewers

Our responses to reviewers' comments are given in blue font beneath each comment.

Reviewer #1

1. I have just a question, linked to the fact that the authors did not explore here the phylogeny of the enzymes they identified. Kinase DrdK is likely to belong to a large family of enzymes. It would be interesting to get some discriminating features of the enzyme in particular contrasting it with 5-methylthioribose kinase. Indeed, in *B. cereus* for example there are two highly similar enzymes, one that can phosphorylate 5-methylthioribose, and the other one that presumably phosphorylates 5-deoxyribose, both belonging to the choline kinase family (PMID: 11545674). This may allow extraction of a signature that would help annotating further genomes.

We agree with this point and have tried to separate 5-methylthioribose kinase and 5-deoxyribose kinase clades to extract signatures. However, the lack of biochemical evidence means that substrate preference must in most cases be inferred from gene clustering patterns. Unfortunately such inferences are compromised by the many cases where (i) the methionine salvage and 5-deoxyribose salvage clusters are together and contain only one kinase gene, (ii) a kinase gene occurs in only one of the two clusters, and (iii) no kinase gene is present in either cluster (Supplementary Fig. 11). Also, our biochemical data (Table 1, Supplementary Figure 10) show that that DrdK acts on both 5-deoxyribose and 5-methylthioribose, indicating that a binary split between kinase clades may be elusive.

2. ...because this work should be widely used by annotation pipelines, I think that there could be some improvement in the nomenclature. I am well aware, for example, that S-adenosylmethionine is commonly abbreviated into SAM. This is however unfortunate, for many reasons. In particular software for automatic retrieval, when using SAM as a keyword will stumble on so many entries that this will become useless. The common abbreviation...AdoMet would help considerably in that matter. The same view holds for several other abbreviations in the paper. DOA could be dAdo (convenient but still somewhat confusing). Drd might be OK (some overlap with *Drosophila* proteins).

DOA has been changed to dAdo throughout the manuscript.

We agree that AdoMet is preferable to SAM, but we use SAM only as part of the term "radical SAM", which occurs far more often in the literature than "radical AdoMet" (95,000 vs. 4,200 Google hits). We therefore prefer to retain SAM.

We chose the abbreviation Drd because it has not been previously used in bacterial genetics and the likelihood of confusion with *Drosophila* proteins is small.

Reviewer #2

Major Comments:

1. However, one of the major issues with the paper is the data in Fig. 3. It worries me that the intercellular concentration of 5-dR has to reach 20-40 mM to see any defect in growth rate with both the Wt or mutant strains of *B. thuringiensis*.

We thank the reviewer for raising this point. We now realize that estimating intracellular concentrations is problematic due to its reliance on values for cell number per OD unit and cell volume that can vary by more than an order of magnitude (PMID: 16874555, 6780526, 24287933). We have therefore now followed the normal practice of expressing metabolite levels on a protein basis in Fig. 3c. The text and Methods have been modified accordingly.

2. I worry that DrdI, DrdK, and DrdA are only “moonlighting” in vitro, and a different set of enzymes are responsible for the removal of 5-dR in vivo. In fact these enzymes seem more effective at removing 5-methylthioribose (Supp Fig. 10). This and the following issues should be addressed before publication.

We agree, and indicate in the Results and the Discussion that moonlighting by methionine salvage enzymes is involved in 5-deoxyribose metabolism in some organisms. However, this seems unlikely to be the case in *B. thuringiensis*, which has separate sets of methionine salvage and 5-deoxyribose metabolism genes (Supplementary Figs. 2, 11).

Moreover, data from a new study indicate that various *drdA* genes are not involved in the metabolism of many common sugars, which is consistent with a specialist function such as 5-deoxyribose salvage. This point is now made in the Results (p. 4) and the new paper is cited (Ref. 17).

3. Online Materials and Methods should contain a section describing how DrdA was prepared in the presence of added metals for ICP analysis. It states in the paper that “a mineral supplement was included in the growth medium,” but is not described in the Methods.

The preparation of the protein for ICP-MS analysis has been clarified in the Methods section.

4. ...the metal dependent assays are briefly described in the Fig. 2 legend, but not in the Methods.

The metal-dependent assays are now described in the Methods.

5. Was the enzyme used from mineral supplement growths plus added metals or just from LB growth and then added metals? Is the same true for the metal analysis?

The DrdA used for *in vitro* assays was His-tagged, isolated from cells grown in unsupplemented LB medium, and purified by Ni-NTA. It was then stripped of metals with EDTA and desalted.

The DrdA used for metal analysis was purified without a His-tag.

This information, previously given only in the Methods, has now been added to the legends of Figure 2 and Supplementary Fig. 6.

6. These two experiments should be congruent. These issues make it difficult to evaluate the metal dependence of DrdA.

The experiments cannot be fully congruent because the tag on the His-tagged protein will itself bind divalent cations, making interpretation of the results very difficult.

7. Supplementary Fig. 6a should be converted to metal/protein monomer. The numbers have little value as presented. In panel b the authors present an EPR spectra of their protein. They state in the paper “ICPMS and EPR spectrometry were used to identify the bound metal in DrdA.” In the methods section, the authors don’t say how much protein was used, but added 1 mM MnCl₂: no other metals are listed. This is a very biased experiment.

We added metal/protein ratio values to Supplementary Fig. 6a as suggested, but retained ppb as this is commonly used to describe metal content. We changed this sentence to: “ICPMS and EPR spectrometry were used to assign and characterize the bound metal in *DrdA*”.

The amount of protein used has been clarified in the Methods section.

The EPR experiment was ‘biased’ toward the manganese complex because this is the metal we identified using ICP-MS. We wanted to go into the EPR analysis with a homogeneous sample, so we prepared the metal bound complex and did the characterization. A sentence has been added in the text to make this clearer.

8. In addition, Mn²⁺ always has a six-line spectrum whether bound or free in solution. Based on the structure of *DrdA*, the Mn is bound by three His residues. These residues should contribute hyperfine coupling to the EPR spectra, which should be present and can be simulated. Therefore the EPR spectra needs better characterization.

Mn(II) indeed shows a six-line spectrum due to the nuclear spin of $I=5/2$ of ⁵⁵Mn. Indeed, it is not always possible to distinguish Mn(II) species just based on their spectra near $g=2$ (around 3500 G at X-band) because of the relative insensitivity of the Mn hyperfine coupling to the Mn environment. This is due to the fact that the ground state of Mn(II) is ⁶S₀. The fine structure of the Mn(II) ion is often the only way to distinguish between aqueous and protein-bound Mn. Aqueous Mn(II) usually has a small fine structure value while it is often much higher in proteins due to the more asymmetric electrostatic environment there. Species with larger fine structure parameter show spectral broadenings and characteristic ‘humps’ at low (and high) fields. The EPR spectrum in the supporting information has been supplemented with a spectrum of just the aqueous Mn(II) in the buffer medium. As one can see, the $g=2$ region is very similar indicating that unbound aqueous Mn(II) is also present in the preparation which is not unexpected. However, additional peaks at low field, particularly near 1200 and 2000 G, indicate Mn(II) with a larger fine structure as expected for a protein-bound complex. The reviewer mentions the effect of coordinating nitrogens on the EPR spectrum which is often observed in Cu(II) complexes. This so-called superhyperfine coupling is very weak for Mn(II) because it is an S-ion and never directly observed by EPR alone. One would need to perform ENDOR or ESEEM experiments to be able to find these comparably small coupling constants. These experiments are beyond the scope of this contribution.

9. Also, the as-isolated protein (without added metals) from the mineral supplement media seems like it would be a better source for the sample. This should be done and would definitively prove the Mn is bound to the enzyme as the structure predicts.

As described above in point 7, the sample from mineral supplement medium is heterogeneous, containing a mixture of metal complexes (Supplementary Fig. 6a). In order to accurately characterize the metal binding with EPR we prepared a homogeneous Mn sample.

Minor Comments:

1. Reference 6 is to an enzyme called ThiC. Although the enzyme produces DOA, it is not a member of the RS superfamily of enzymes. The authors state this mistake in the sentence “If not removed, DOA can reach toxic levels that inhibit radical SAM enzymes themselves” by referencing ThiC as an RS enzyme.

ThiC is classified as a unique member of the radical SAM family. See:

Chatterjee *et al.* Reconstitution of ThiC in thiamine pyrimidine biosynthesis expands the radical SAM superfamily. *Nat. Chem. Biol.* **4**, 758–765 (2008)

Chatterjee *et al.* A “radical dance” in thiamin biosynthesis: mechanistic analysis of the bacterial hydroxymethylpyrimidine phosphate synthase. *Angew. Chem. Int. Ed.* **49**, 8653-6 (2010)

2. First sentence of bottom paragraph on Page 2: "It has been proposed, but not proved, that the archaeon" seems redundant. "But not proved" should be removed.

"But not proved" has been deleted.

3. It should be discussed more about what happens in species like *E. coli*. *E. coli* has 20 RS enzymes and several of them have been shown to be particularly sensitive to inhibition by DOA both *in vivo* and *in vitro*.

The fate of 5-deoxyribose in *E. coli* is unknown. We suggest in the Discussion that, like 5-methylthio-ribose, it may be excreted into the medium. We prefer not to speculate further at this time.